# E-Learning Performance Prediction: Mining the Feature Space of Effective Learning Behavior

**DOI:** 10.3390/e24050722

**Published:** 2022-05-19

**Authors:** Feiyue Qiu, Lijia Zhu, Guodao Zhang, Xin Sheng, Mingtao Ye, Qifeng Xiang, Ping-Kuo Chen

**Affiliations:** 1College of Education, Zhejiang University of Technology, Hangzhou 310023, China; qfy@zjut.edu.cn (F.Q.); 2112008046@zjut.edu.cn (L.Z.); 2111908095@zjut.edu.cn (X.S.); 2112008049@zjut.edu.cn (Q.X.); 2Department of Digital Media Technology, Hangzhou Dianzi University, Hangzhou 310018, China; 3College of Computer Science and Technology, Zhejiang University of Technology, Hangzhou 310023, China; 2112012101@zjut.edu.cn; 4Business School, Shantou University, Shantou 515000, China

**Keywords:** E-learning performance, E-learning behavior classification, feature space, machine learning, feature fusion

## Abstract

Learning analysis provides a new opportunity for the development of online education, and has received extensive attention from scholars at home and abroad. How to use data and models to predict learners’ academic success or failure and give teaching feedback in a timely manner is a core problem in the field of learning analytics. At present, many scholars use key learning behaviors to improve the prediction effect by exploring the implicit relationship between learning behavior data and grades. At the same time, it is very important to explore the association between categories and prediction effects in learning behavior classification. This paper proposes a self-adaptive feature fusion strategy based on learning behavior classification, aiming to mine the effective E-learning behavior feature space and further improve the performance of the learning performance prediction model. First, a behavior classification model (E-learning Behavior Classification Model, EBC Model) based on interaction objects and learning process is constructed; second, the feature space is preliminarily reduced by entropy weight method and variance filtering method; finally, combined with EBC Model and a self-adaptive feature fusion strategy to build a learning performance predictor. The experiment uses the British Open University Learning Analysis Dataset (OULAD). Through the experimental analysis, an effective feature space is obtained, that is, the basic interactive behavior (BI) and knowledge interaction behavior (KI) of learning behavior category has the strongest correlation with learning performance.And it is proved that the self-adaptive feature fusion strategy proposed in this paper can effectively improve the performance of the learning performance predictor, and the performance index of accuracy(ACC), F1-score(F1) and kappa(K) reach 98.44%, 0.9893, 0.9600. This study constructs E-learning performance predictors and mines the effective feature space from a new perspective, and provides some auxiliary references for online learners and managers.

## 1. Introduction

In the last decade, with the concept of ”lifelong learning” and ”individualized learning”, E-learning was widely developed worldwide due to its characteristics of massive information, strong interaction, wide coverage, and no time and space restrictions [1], so it provides convenience for learners. However, at the same time, compared with traditional classrooms, in electronic learning, it is difficult for teachers to directly obtain student performance in the course and provide timely feedback. They usually get information about the failure of the course after the course is over. This leads to poor learning effects and lower students’ confidence. Undergraduates may choose to drop out or postpone their studies in these situations. One way to detect whether students will perform poorly in a course is to make early predictions of their grades [2]. At present, many scholars are committed to the field of predictive modeling learning analysis and educational data mining. Learning behavior data decomposes the originally complex learning behavior into operable and obtainable data indicators [3], and deepens our understanding of learning process through the analysis of learner-related data [4]. The purpose of learning performance prediction is to predict and understand the academic performance of students in the learning trajectory, help teachers to comprehensively understand students’ academic conditions, and implement targeted intervention plans based on the predicted results to improve the learning experience of learners [5]. However, using this method also faces multiple challenges, especially how to obtain, process and use data to build learning behavior models [6]. It can be seen that predicting learners’ academic performance has become a key issue in the field of learning analysis. When mining the prediction indicators of learning achievement and academic risk, it focuses on demographic data, behavioral data, and students’ past performance [7]. As demographic data involves the privacy of students, the deletion of some information may affect the availability of data and the accuracy of the results, so there are certain limitations. At present, there are many related studies published in the field of student test score analysis, the purpose is to predict the ”high risk” of students withdrawing from a particular course. Although good results have been achieved, there are still some limitations. First, many studies tend to choose multi-dimensional, fine-grained behavioral data [8,9,10] and have achieved good results. However, few people consider isolated learning behaviors as a whole, ignoring the internal connection of learning behaviors, or only considering single or several behavioral data are used as prediction indicators [11,12], and it is difficult to highlight the impact of core behaviors on learning performance. Second, when using multi-dimensional data, irrelevant variables will affect the generalization ability of the model and increase the computational cost of the model. With the huge increase in E-learning platform data, these problems have gradually become prominent. Third, the index input method for constructing the model is single, and different types of features may affect the accuracy of prediction [13]. However, few people consider the combination of different types of student-related features to determine the best combination of model input data. Fourth, using traditional preprocessing methods to construct learning performance predictors also has some problems, such as prediction accuracy and computation time. The high accuracy and low time-consuming rate of performance prediction can effectively support teachers to track the performance of each student, and can further guarantee the quality of online learning.

Addressing these limitations can help discover effective ways to improve learning analytics research in higher education. To this end, this paper constructs a learning performance predictor from a new perspective, that is, based on the E- learning behavior classification model (EBC Model), the E-learning behavior is divided into four categories, and different combinations of the four categories of behavior are used as input variables. Prediction analysis and performance comparison of machine learning models to mine the effective behavioral feature space. Then, the entropy weight method, variance filtering method and self-adaptive feature fusion strategy are combined in the data preprocessing stage to further improve the performance and efficiency of the predictor. The contributions of this work can be summarized as follows:

1. On the basis of teaching theory, construct an E-learning behavior classification model (EBC-Model) based on E-learning process and interactive objects, and provide a reference for educational researchers to standardize the classification of behavior classes.

2. The class feature space is constructed based on the EBC Model. The feature space is the set of behavior class features through combination and transformation. We use the variance filtering method and the entropy weight method to filter the behavior features to reduce the influence of low expressive features on the predictor. Then, a variety of algorithms are used to analyze the feature subset with higher prediction effect in the feature space, and then divide it into an effective feature space. It explores the feature space that is more related to learning performance from a new perspective, and considers the internal correlation between behavioral features, which provides an auxiliary reference for educators to focus on what behavioral features.

3. We propose a learning performance predictor based on a self-adaptive feature fusion strategy. First, the features are visually clustered and two fusion strategies of different clusters are observed. Then, the Euclidean distance between the students and the clusters is calculated, and the feature fusion strategy is adaptively selected. Finally, it is verified by experiments that the strategy can effectively improve the performance and efficiency of the predictor.

This article is composed of 6 summaries, and the rest of the content is as follows: Section 2 focuses on related research, focusing on the correlation between electronic learning behavior and performance, E-learning behavior classification and E-learning performance prediction indicators, data processing methods and prediction model; Section 3 explains our proposed method and process; Section 4 describes two experimental designs, which are the experimental designs for mining the effective E-learning behavior feature space and verifying the effectiveness of feature fusion; Section 5 analyzes the results of the two experiments separately; Section 6 gives conclusions and future directions.

## 2. Related Work

### 2.1. The Correlation between E-Learning Behavior and Performance

With the continuous advancement of educational information construction, electronic learning has achieved a blowout development, and the analysis of learning behavior is the focus of the field of learning analysis. Electronic learning behavior has been proved to be closely related to learning performance [14,15]. Current research mainly focuses on the online participation of the entire online learning system. Bolliger et al. [16] found that there is an inevitable connection between student engagement, participation awareness and learning results. Through sampling, Shen et al. [15] gradually regressed and found that behaviors such as homework completion rate and video completion rate have an important positive impact on the final online learning effect. Zheng et al. [17] found that the number of logins is positively correlated with academic performance, and project-based assignments and high-level knowledge activities are beneficial to learning results. Qureshi et al. [18] used structural equation model evaluation to find that learners’ social participation and social interaction affect students’ enthusiasm for collaborative learning and participation, thereby affecting their academic performance. Mehmet et al. [19] found that the interactive behavior of the learner and the instructional learning dashboard is significantly related to the learner’s academic performance. In addition, existing studies have also used cluster analysis to analyze the relationship between student interaction patterns and learning achievement. Rebeca et al. [20] divides students into two task-oriented groups (social or individual-centered) and two non- Task-oriented groups (procrastinators or non-procrastinators) for better targeted teaching. Electronic learning behavior data is used in many areas of learning analysis, including learner modeling, learning performance prediction, etc. When exploring the relationship between electronic learning behaviors and learning performance, research focuses on single or independent learning behaviors, and the potential between behaviors. The underlying rules of the association structure are not significant. Therefore, some scholars have begun to explore the internal connections between behaviors to express the subtle and complex logical relationships in learning, and apply them to learning analysis.

### 2.2. The Development of E-Learning Behavior Classification

Regarding the study of learning behavior classification, many scholars have expounded from the perspective of learning process and interactive objects. Moore [21] defines the first interaction classification framework from the perspective of interactive senders and receivers, including three basic interactive behaviors in Electronic learning: student-student interaction, student–teacher interaction, and student–content interaction. The interaction between students and students occurs in independent learning or in a group environment. The interaction between teachers and students lies in the timely guidance and intervention of teachers in learning to promote students’ deep learning. The interaction between students and content is defined as the intellectual interaction between students and content to change learners’ understanding, viewpoints and cognitive structure. Hillman et al. [22] further proposed the fourth type of learning behavior-student-student interface interaction behavior, that is, the behavior of students using electronic tools and navigation tools. Based on the lack of explanations for teaching activities in the above two classifications, and in order to obtain a more general classification, Hirumi [23] divides learning behaviors into learner-self interaction, learner-human and non-human interaction, learner-teaching interactive. Sun [24] considers that the cognitive structure of students is dynamically changing in the learning process, and divides the learner’s E-learning behavior process into four stages: learning occurrence stage, knowledge acquisition stage, interactive reflection stage, and learning consolidation stage. Wu et al. [25] divided E-learning behaviors into independent learning behaviors, system interaction behaviors, resource interaction behaviors, and social interaction behaviors based on the interrelationships between learners and the three basic elements of the electronic learning space (learning systems, learning resources, and people). There are four types of interactive behaviors. Wu et al. [10] divides E-learning behaviors into four categories according to the interactive subjects, namely student-student interaction, student–teacher interaction, student–content interaction, and student–system interaction. To sum up, current studies only consider learners’ behaviors from a single perspective of learning objects or learning processes, and do not combine complex learning processes with learners’ interactive behaviors, so as to show the coordination and unity of dynamic changes in learning processes and interactive behaviors.

There is a significant correlation between E-learning behavior and learning performance. Using learning behavior to predict performance is a current research hotspot. However, the use of multi-dimensional data may affect the generalization ability of the prediction model and there is a high computational cost. Constructing learning behavior classification rules is conducive to mining the potential associations between behaviors, and provides a standard for dividing learning behaviors, which is useful for promoting the standardization and lightweight of E-learning performance prediction models are of great significance.

### 2.3. E-Learning Performance Prediction

#### 2.3.1. Prediction Indicators of E-Learning Performance

Learning performance indicators are generally divided into two types of commonly used prediction indicators, propensity indicators and behavioral performance indicators. Tendency indicators refer to the students’ own attributes (including gender, age, race, etc.) and past experience [26]. After entering the learning environment, they will not change with the progress of the learning process, so they are also called static indicators [27]. Marbouti et al. [28] use previous academic performance, homework and test scores as prediction indicators to find the best prediction model. Musso et al. [5] use students’ demographic background factors, learning strategies, coping strategies, cognitive factors, social support, and self-satisfaction. The construction of a prediction model found that background information has the greatest predictive weight on whether a student will drop out. These studies try to find the relationship between student attributes and past performance and current learning performance. However, it has not considered that many tendency indicators are not under the control of students and teachers. In addition, demographic and psychometric data involve privacy and data availability is limited [29]. Therefore, more research should focus on using student activity data during the course.

Behavioral performance indicators refer to dynamic indicators in the learning process, including searching for resources, watching videos, and forum discussions. As learning progresses, the role of static indicators will gradually decay, and learning performance indicators become the core prediction indicators. Current research focuses on the use of learning analysis methods to explore the core learning behaviors that predict learning performance. Some scholars analyzed the prediction indicators of the model from the perspective of E-learning behavior classification. Li et al. [14] used 41 behaviors input measurement indicators to perform regression analysis, and divided the single behavior data into the behavior input classification. Finally, he found that homework engagement, active communication and knowledge page browsing had higher correlation with course performance. Wu et al. [25] used online course login times, online duration, resource browsing times and other behaviors to construct six classifiers such as random forest and J48 decision tree, and proposed the structure model of electronic learning space to further divide behavioral indicators into categories. Finally, he concluded that independent learning behavior has a strong predictive effect on academic performance. Some scholars directly use the feature data in the E-learning platform to mine the different effects of specific learning behaviors on the prediction model. For example, Shen et al. [15] used the learning behavior data obtained on the MOOC platform to conduct sampling regression and found that homework completion ratio and video completion ratio had an important positive impact on learning effect. Zacharis [8] used 29 variables in the blended learning courses supported by LMS on Moodle to find that reading and publishing information, content creation, resource browsing, and completion of quizzes are more important predictors through multiple regression methods. Macfadyen et al. [30] believe that forum posts, emails, and quizzes are important predictors of students’ final grades.Brinton et al. [11] constructed predictors of learning performance by using students’ video-watching behavior and test performance on MOOC platform. Yu et al. [9] used behaviors such as interaction with peers, download times and access to resources as predictors of academic performance. There are also studies that use students’ homework submission behavior to predict students with learning difficulties and judge the tendency of procrastination through students’ submission behavior to predict performance [31].

In general, many domestic and foreign scholars focus on completing quizzes, resource browsing, content creation, resource downloading, course visiting, watching videos, submitting homework and other behavioral indicators to predict and improve students’ E-learning performance. In addition, some scholars consider the correlation between behavior and academic performance from the perspective of behavior classification, but few studies use behavior categories as a predictor to build a predictor of learning performance.

#### 2.3.2. Feature Engineering

Before applying student-related data to learning analysis, you must first consider the availability of the data. Any type of error information stored in the basic data set for deploying learning analysis may have a huge negative impact on the accuracy of the model [32]. Therefore, it is particularly important to fully obtain and accurately process the relevant data and student characteristics that affect the prediction performance. All data operations in the process of transforming the original data into the training data of the model are collectively referred to as feature engineering. Feature engineering is generally considered to include three parts: data preprocessing, feature selection, and feature extraction.

Data preprocessing is the first and necessary step of feature engineering, including cleaning and standardizing the original data with different dimensions, information redundancy, anomalies, and missing data. Feature selection is used to screen meaningful feature input training models [33]. An effective feature selection process can remove redundant data and eliminate feature data that has a low impact on prediction, thus greatly simplifying the structure of the prediction model, reducing the operation cost of the model, and ultimately improving the performance of the model [34]. Variance filtering method, chi-square test, the entropy weight method, correlation coefficient method, and information gain method are usually used to remove redundant features [28]. Wu [10] and Hu [35] et al. used correlation coefficient method and information gain method to extract meaningful feature variables, and the results confirmed that the prediction performance of the model has been greatly improved after feature selection. There are also some scholars who have made improvements to traditional feature selection methods. To solve the problem that traditional machine learning classification algorithms have low accuracy on unbalanced data sets, Chen et al. [36] has proposed a feature selection method ($\mathrm{RFG}-\chi^{2}$) based on the random forest Gini index and chi-square test for optimal feature subset and applied to support vector machine algorithm model. Wang et al. [37] improved the feature selection method of the random forest (RF) standard algorithm, and proposed an RF-based infrared spectrum modeling method and compared it with partial least squares regression (PLS), support vector machine (SVM) and standard RF algorithms. The results show that the improved RF algorithm can improve the accuracy of the infrared spectrum model and reduce the complexity of the model. Wang et al. [38] proposed automatic width search and attention search adjustment methods to further accelerate feature selection based on random wrappers.

Unlike feature selection that selects a better subset from the original feature data set, feature extraction generates new attributes through the relationship between attributes and changes the original feature space [39]. Common methods include principal component analysis (PCA) [40] and independent component analysis (ICA) [41], linear discriminant analysis (LDA). Considering that a single type of index may be one-sided, some studies have also tried to use multi-dimensional features to complete academic predictions, but more and more studies have confirmed that when machine learning modeling, multi-feature data may bring lower prediction accuracy [42]; therefore, it is necessary to use feature extraction methods to reduce feature dimensions to improve the prediction effect and reduce the computational cost of the model [43].

Different machine learning techniques and algorithms can provide satisfactory results, and the difference in performance may depend on the preprocessing process of the available input data. Although there are studies that optimize data processing methods, they are hardly used for learning prediction tasks. Therefore, in order to improve the performance of the predictor, this paper uses a variety of feature engineering methods to process the learning behavior feature data.

#### 2.3.3. Prediction Model

With the continuous development of emerging disciplines such as learning analysis and educational data mining, the use of computing tools and methods to collect and analyze data has attracted widespread attention. Among them, the use of machine learning technology to process large and complex data in the educational environment for classification tasks can be used for prediction analysis [44], such as student test score prediction. Tomasevic et al. [7] divide the methods of constructing prediction models into similarity-based, model-based and probability-based methods.

Similarity-based methods identify similar patterns (for example, students with similar past academic performance). The typical similarity-based method is K-Nearest Neighbour (KNN). The test samples are classified according to the majority vote of their neighbors, and the samples are assigned to the most common class among their nearest neighbors accordingly. However, KNN needs to calculate a large number of sample distances, and is not suitable for situations where sample data is not balanced. Therefore, a KNN algorithm based on distance weighting is proposed, which assigns weights to k nearest neighbor data, and increases the weight when the distance between the test sample and the designated neighbor decreases [45].

The model-based method is driven by the implicit correlation estimation between the input learning data. SVM is one of the typical representatives of the model-based method. It is mainly used to solve the data classification problem in the field of pattern recognition and belongs to a kind of the supervised learning algorithm. SVM can be regarded as the problem of finding the effective separation hyperplane for separating data [46], and can be used effectively for linear classification using the so-called kernel technique. When the relationship between the specified category and the input features (that is, the output and input variables) is nonlinear, the radial basis function (RBF) is usually combined as the kernel function. Softmax classifier is also a model-based method, which is the extension of Logistic Regression model on multiple classification problems. In multi-classification problems, classification labels can take more than two values.

Finally, the third is probability method that uses the probability of the feature distribution of the input data set to classify. One of the widely used probabilistic classification methods is the Bayesian classifier. Bayesian decision theory considers how to select the effective category label based on these probabilities and misjudgment losses. Bayes is a fast, highly scalable algorithm that can be used for binary and multicast classification [47]. In addition, Decision Tree (DT) is also a probability-based classification method. DT uses a series of rules to classify data. It has many advantages, such as strong interpretability, fast computational speed, high accuracy, insensitivity to missing values, and does not require any domain knowledge and parameter assumptions. However, when dealing with data with strong feature correlation or continuity fields, it is weak.

All three advanced methods and their typical techniques were used to find, “high risk” of students who leave high school before finishing and carries on the evaluation of examination results, in order to solve classification and regression. In order to comprehensively analyze the effective subsets of the feature space and verify the improvement of the predictor performance, all the above classification models are selected for our work in this paper.

## 3. Methods

### 3.1. Problem Definition

Through measurement, collection and analysis of learning behavior process data to achieve learning performance prediction, after the beginning of the course can be based on the predicted results for students to carry out certain teaching intervention, but also play a role of supervision and early warning to students, reduce the risk of course failure. Mining excellent indicators for predicting academic performance has always been the focus of learning analysis. This study tries to explore which combination of behavioral categories can better predict academic performance from the perspective of learning behavior classification, and then excavates the effective behavioral feature space. We assume that there are *M* students and that each student has an E-learning behavior set *A*, $A\left\{a_{1},a_{2},\ldots ,a_{N}\right\}$, where $a_{n}$ is the label of E-learning behavior, n=1,2,…,N, *N* is the number of E-learning behaviors. Set the corresponding standardized E-learning behavior $A^{\prime }\left\{a_{1}^{\prime },a_{2}^{\prime },\ldots ,a_{N}^{\prime }\right\}$ according to $A\left\{a_{1},a_{2},\ldots ,a_{N}\right\}$. Define the set of characteristic values of E-learning behavior $S\left\{s_{1},s_{2},\ldots ,s_{N}\right\}$, $s_{n}$ represents the overall eigenvalue of the nth learning behavior.The entropy weight method and variance filtering method are used to reduce the dimensionality of the set of eigenvalues *S*. Then, the E-learning classification model (EBC Model) is used to divide learning behavior labels $a_{n}^{\prime }$ into different behavior classes. After the classification of behavior categories, $N_{b}$ behavior classes will be generated, and different behavior classes contain different learning behavior labels $a_{n}^{\prime }$. Define *E* as the set of each behavior class, such as $E_{1}\left\{a_{1}^{\prime },a_{3}^{\prime },\ldots ,a_{i}^{\prime }\right\}$, $a_{i}^{\prime }$ represents the E-learning behavior that meets the standard. In this paper, the feature fusion strategy is adopted to improve the performance of the predictor. $s_{E_{1}}$ is defined as the eigenvalue of the behavior class $E_{1}$, and the eigenvalue set of the behavior class $S_{E}\left\{s_{E_{1}},s_{E_{2}},\ldots ,s_{E_{N}}\right\}$. Table 1 shows the notation interpretation.

### 3.2. Method

In this section, we will introduce the process of obtaining a new learning behavior classification based on the EBC Model, and explain the process of preprocessing, feature selection, feature fusion and model training based on the behavior classification data. Figure 1 shows the main process of the work of this paper, including five core links: (1) Data preprocessing: clean and standardize the original data of the data set. (2) Feature selection: select the more relevant learning behaviors to reduce data redundancy and feature dimensions. (3) Learning behavior classification: classify the original learning behavior according to the E-learning classification model (EBC Model). (4) Feature fusion: a self-adaptive feature fusion strategy is proposed to obtain categorical feature values. (5) Model Training: construct a learning performance predictor based on a variety of machine learning algorithms to explore the effective behavior feature space, and a self-adaptive feature fusion strategy is used to improve the performance of the prediction model.

#### 3.2.1. Behavior Classification—E-Learning Behavior Classification Model (EBC Model)

The E-learning environment mainly includes three basic elements: learning system, learning resources and learning community (teachers and peers). The interactive behavior of learners with the learning community, learning system, and learning resources is called E-learning behavior. Constructing an E-learning behavior classification model is of great significance to the collection and measurement of E-learning behavior, and the measurement of E-learning behavior directly affects the analysis of learning behavior. Therefore, this article proposes the E-learning Behavior Classification Model (EBC Model), which provides a reference for standardized classification behavior.

As shown in Figure 2, in order to reflect the process of E-learning and the interaction between learners and different objects, this article is based on the classification of Sun [24] and Wu [25], and divides the interaction between learners and interactive objects in the process of E-learning. For basic interaction behavior, knowledge interaction behavior, community interaction behavior and self-interaction behavior, each type of learning behavior corresponds to a different learning stage. The following are the four learning behavior categories classified according to the EBC Model:

Basic interactive behavior (BI): refers to the interactive behavior of learners and the E-learning system, which occurs at the stage of learning, and reflects the degree of learning input. It generally includes behaviors such as logging in to the learning platform, visiting the main page of the course, and visiting the course activity interface. Knowledge interaction behavior (KI): refers to the interaction behavior between learners and system learning resources, which occurs in the knowledge acquisition stage, reflecting learners’ utilization and preference of learning resources, generally including searching, accessing, downloading resources, watching videos, etc. behavior. Community interaction behavior (CI): refers to the interactive behavior of learners, teachers, and peers. It occurs in the interactive reflection stage and reflects the learner’s degree of learning interaction and reflection. It is specifically manifested as participation in seminars, forum discussions, collaborative communication and other behaviors. Self-interaction behavior (SI): refers to the behavior of learners to process, encode, and memorize knowledge. It reflects the achievement of learners’ goals. It occurs in the stage of learning consolidation and runs through the entire process of learners’ self-construction of knowledge. The main explicit behaviors include after-school testing and homework submission.

#### 3.2.2. Data Preprocessing

(1) Data cleaning.

Due to various reasons, there may be duplicate records or duplicate fields, outliers, and missing values in behavioral data. Duplicate items need to be removed, and then the abnormal behavior index value of each case is replaced with missing values, and finally the respective missing values are filled in according to the average value of other students on the corresponding behavior variables.

(2) Standardization.

The E-learning platform records multi-dimensional E-learning behaviors, and the dimensions of different behavior variables are also different, resulting in large differences between actual values, which do not have the meaning of direct comparison. In order to facilitate subsequent analysis, this study uniformly converts the values of n preset learning behaviors into standard scores (Z-Score), discards the case of not taking the exam (periodical test or final exam), and replaces the remaining student behavior related data Scale and normalize. Z-Score unifies data standards, improves data comparability, and weakens data interpretability.

Define the original E-learning behavior data $p_{mn}$, the standard learning behavior data $p_{mn}^{\prime }$ where *m* represents the student user tag, such as $p_{21}$ represents the first type of learning behavior data of the second student recorded by the E-learning platform. The calculation formula of the standard E-learning behavior value $p_{mn}^{\prime }$ is as follows:(1)$$p_{mn}^{\prime }=\frac{p_{mn}-\mu_{a_{n}}}{\sigma_{a_{n}}}$$

Among them, $a_{n}$ is the E-learning behavior label, $\mu_{a_{n}}$ represents the average value of the nth E-learning behavior data, and $\sigma_{a_{n}}$ represents the variance of the *n*th E-learning behavior data.

#### 3.2.3. Feature Selection

The feature selection method is to select high-quality features from the original features, delete irrelevant and redundant features, and realize the process of reducing the feature dimension of the data set. It can improve classification efficiency and accuracy, and has the effect of denoising and preventing over-fitting. It is an important means to improve the performance of machine learning algorithms.

(1) The entropy weight method determines the objective weight based on the variability of the index. Generally speaking, the smaller the information entropy of an index, the greater the degree of variation of the representative index, and the more information it provides, so it is used in feature selection. The greater the effect, the greater the weight, and vice versa.

First, calculate the proportion of the behavior data of the *i*th student in the *j*th E-learning behavior, the calculation formula of $Q_{ij}$ is:(2)$$Q_{ij}=\frac{p_{ij}}{\sum_{i=1}^{m}p_{ij}}$$
where *m* represents *m* students, $p_{ij}$ is the standardized E-learning behavior data. Then, calculate the information entropy $e_{j}$ of E-learning behavior,
(3)$$e_{j}=-\ln{\left(m\right)}^{-1}\sum_{i=1}^{m}Q_{ij}\ln Q_{ij}$$
and the weight of each behavior $\omega_{j}$ is calculated through the information entropy,
(4)$$\omega_{j}=\frac{1-e_{j}}{\sum_{j=1}^{n}\left(1-e_{j}\right)}$$
where *n* is the number of E-learning behaviors. Finally, calculate the comprehensive score $F_{i}$ of each E-learning behavior,
(5)$$F_{i}=\sum_{j=1}^{n}\omega_{j}k_{ij}$$
where $k_{ij}$ is the calculated relative deviation fuzzy matrix value.

This article selects the top eight E-learning behaviors with comprehensive scores, and the experimental results are basically consistent with the results obtained by the variance filtering method.

(2) Variance Threshold filters features through the variance of the features themselves. For example, if the variance of a feature is small, it means that the samples have basically no difference in this feature, so this feature has little effect on the distinction of samples. Threshold represents the threshold of variance, that is, discarding all features with variance less than Threshold. The calculation formula of the characteristic value $S_{n}$ of the nth learning behavior is as follows:(6)$$S_{n}=\frac{\sum_{1}^{m}{\left(p_{mn}^{\prime }-\mu_{a_{n}^{\prime }}\right)}^{2}}{m}$$
Among them, $\mu_{a_{n}^{\prime }}$ represents the average value of the standardized E-learning behavior data $a_{n}$, m=1,⋯,M. Compare the elements in the traversal learning behavior feature value set S with the variance threshold, and take the E-learning behavior feature value greater than the threshold.

#### 3.2.4. Feature Fusion

In order to solve the problem of low accuracy and high computational cost of traditional machine learning classification algorithms in the face of huge data sets, we use feature fusion methods to further improve the performance of the predictor.

First, the behaviors are divided into different E-learning behavior clusters according to the EBC Model, and *n* E-learning behavior clusters are generated. Any type of E-learning behavior cluster contains different E-learning behaviors, such as $E_{n}\left\{a_{1}^{\prime },a_{3}^{\prime },\ldots ,a_{i}^{\prime }\right\}$ etc. Among them, $a_{i}^{\prime }$ represents the E-learning behavior that complies with the $E_{1}$ standard.

Then, feature fusion is performed on each E-learning behavior cluster to obtain the corresponding E-learning behavior category feature value. Taking $E_{1}$ as an example, the calculation formula of its category eigenvalue is as follows:(7)$$S_{E_{n}}=\lambda \cdot \max\left\{s_{1},s_{2},\ldots \ldots ,s_{i}\right\}+\left(1-\lambda \right)\cdot \frac{\sum_{1}^{i}s_{i}}{i},\lambda =\left\{\begin{array}{c} 0,\;\mathrm{Pass}\;\\ 1,\;\mathrm{Fail}\; \end{array}\right.$$
Among them, $s_{i}$ represents the feature value of $a_{i}^{\prime }$ behavior, and $S_{E_{n}}$ represents the E-learning behavior category feature value of $E_{1}$. λ represents the fusion mode of the student behavior class.

The choice of λ is obtained by comparing the similarity of clusters. This paper classifies students based on behavioral characteristics by K-means clustering. The key step in K-means is to find the number of clusters with the best classification effect, namely K value. First, calculate CH (Calinski-Harabaz) values at different K values, Calinski-Harbasz Score calculates the score of the clustering results by evaluating the between-class variance and within-class variance. It can be seen from Figure 3 that the CH value decreases as the K value decreases. Therefore, the optimal K value should be 2.

Finally, principal component analysis (PCA) is performed on the characteristics of students’ E-learning behaviors to reduce the dimension of students’ E-learning behavior characteristics to two dimensions, and then perform visual analysis. Then, as can be seen from Figure 4, it is found that the difference of the distance of the student features of the red class is small, and the maximum value should be adopted for better fusion effect, that is, λ = 1; the difference of the distance of the student features of the yellow class is large, and the mean value should be adopted to achieve a better fusion effect, that is, λ = 0.

In the clustering process, first define the eigenvectors of the behavior feature set of students $U\left\{s_{1},s_{2},\ldots ,s_{i}\right\}$. Regarding how to measure the distance between students, we choose the Euclidean distance as the distance calculation formula, as follows:(8)$$\mathrm{dist}\left(s_{i},s_{j}\right)=\sqrt{\sum_{k=1}^{M}{\left(c_{k}^{i}-c_{k}^{j}\right)}^{2}},\left(s_{i},s_{j}\in U\right)$$
Among them, $\mathrm{dist}\left(u_{i},u_{j}\right)$ represents the Euclidean distance between students $u_{i}$ and $u_{j}$, and $C_{k}^{i}$ represents the behavioral feature $c_{k}$ of learner $u_{i}$. The process of K-means is to randomly select K learners as cluster centers and iterate until the centers do not change. In each iteration, the distance from each learner to the cluster center is calculated and the learners are divided into the nearest clusters, and the cluster center is selected as the cluster center point for the next iteration for each cluster.

After t iterations, the learner is classified into K learner clusters by the K-means algorithm, denoted as $\left\{P_{1},P_{2},\ldots ,P_{K}\right\}$, where $P_{i}$ represents the ith learner cluster. In this paper, K = 2 is obtained through the analysis of Figure 3 and Figure 4.

To compare the clustering similarity, define the predicted learner $u_{t}$, calculate the Euclidean distance between the center point of $u_{t}$ and $P_{K}$, and divide it into the cluster with the closest distance. $P_{S_{i}}$ indicates which cluster $S_{i}$ belongs to. The formula is as follows:(9)$$Ps_{i}=\left\{\mathrm{dist}\left(s_{i},P_{i}\right),\mathrm{dist}\left(s_{i},P_{i}\right)\right\},i\in \left[1,2\right]$$
The final value of λ is determined according to the value of $P_{S_{i}}$. If $P_{S_{i}}$ belongs to the red class in Figure 4, then λ = 1, and vice versa.

#### 3.2.5. Model Training

Choose from seven classic machine learning methods in model training, including SVC (R), SVC (L), Bayes, KNN (U), KNN (D), DT and Softmax. $S_{E}$ is used as the training input data, mining the effective feature space and feature fusion are used to improve the training effect.

## 4. Experimental Design

This experiment is based on our proposed E-learning classification model (EBC Model) to construct a subset of feature behavior classes as the input variables of the model. By evaluating the performance of machine learning algorithms, comparing the prediction performance of different feature subsets to mine the effective behavior feature space, and further use feature fusion data processing methods to improve prediction performance. Data mining and machine learning techniques can be applied to various learning analysis tasks. In the case of student test score prediction, advanced classification methods are used to analyze the “high risk” or “low risk” of students dropping out or failing the final exam. The machine learning methods used include SVC (R), SVC (L), Bayes, KNN (U), KNN (D), DT and Softmax. We design three sets of experiments. Experimental group one uses un-dimensioned feature data to train the predictor, experimental group two uses the feature data after dimensionality reduction to train the predictor, and experimental group three uses the feature data after dimensionality reduction and feature fusion strategy to train the predictor. In terms of performance evaluation indicators, we choose accuracy (ACC), F1-score (F1) and kappa (K) and computation time (Time) as quantitative indicators.

We use Lenovo xiaoxin 14pro Laptop to build the experimental environment, the CPU model is Intel Core i5 1135G7, the memory is 16G, the Windows 10 operating system, and the Jupyter lab software are used for the experiment.

### 4.1. Data Source

Before comparing the effect of learning performance prediction on different input combinations, it is first necessary to determine a suitable data set containing different E-learning behavior data. The Open University Learning Analytics Dataset (OULAD) used in this article to construct the research object is one of the most comprehensive and diverse international public data sets related to student learning. The data set includes 7 course modules from AAA to GGG, 22 courses in total, demographic data of 32,593 students and interaction data between students and VLE. The development of this data set has made a significant contribution to supporting research in the field of learning analysis, through the collection and analysis of learner data to optimize learning resources and provide personalized feedback. This experiment selects the DDD subject of STEM courses in OULAD, and uses E-learning behavior data of 6272 learners who study DDD courses to train the learning performance predictor. According to EBC Model, E-learning behaviors are divided into their respective behavior categories, as shown in Table 2.

### 4.2. Mining the Feature Space of Effective E-Learning Behavior

#### 4.2.1. Experimental Program

Construct multiple predictors by controlling the input variables of the training data, and enumerate 15 combinations of class eigenvalues. Experimental group one uses non-dimension-reduced class eigenvalues to train the predictor, and experimental group two uses dimensionality-reduced class eigenvalues to train the predictor. Analyze the performance of experimental group one and experimental group two through evaluation indicators (accuracy (ACC), F1-score (F1) and kappa (K), and mine the effective feature space.

#### 4.2.2. Feature Selection

In this paper, the entropy weight method and the variance filtering method are used to analyze the behavioral characteristics, as shown in Table 3, and the results obtained by the two methods are basically the same. In order to ensure the diversity of features, at least one feature is retained under each behavior class, and each behavior class is compressed as much as possible. Therefore, we selected 8 characteristics as our research objectives, excluding behavioral data with low correlation with learning performance. The feature data of the E-learning behavior after feature selection is shown in Table 3.

#### 4.2.3. Construct Input Variables for Training Data

In order to explore the effective learning behavior feature space and explore which of the basic interaction behaviors, knowledge interaction behaviors, community interaction behaviors, and self-interaction behaviors are good for predicting students’ academic performance, we use the exhaustive method to divide the four categories of the EBC Model made 15 combinations to construct feature sets, namely F0-F14 (using the E-learning behavior data after feature reduction selection), and use them as input variables to construct predictors, and comprehensively compared the performance indicators of each classification predictor (accuracy (ACC), F1-score (F1), kappa (K), mining the best combination of behavior category feature data. Table 4 shows the details of the complete behavior category combination feature data.

### 4.3. Validation of Feature Fusion

#### 4.3.1. Experimental Program

In order to further improve the performance of the predictor, we design experimental group three to perform feature fusion on the class feature values after feature selection, and then train the predictor. By comparing the predictor performance indicators (accuracy (ACC), F1-score (F1), kappa (K) and computation time (Time)) of experimental group three and experimental group two, it is judged whether the feature fusion strategy has an improvement effect.

#### 4.3.2. Feature Fusion

As shown in Table 4, taking F10 as an example, the input variables of experimental group two have 7 eigenvalues H, S, W, R, U, T, F, in experimental group 3, we perform eigenvalue fusion, according to Equation (Equation 7), the input variable is fused from 7 features into BI, KI, CI. According to the above rules, a predictor for a combination of 14 other categories is constructed.

### 4.4. Realization and Evaluation of the Predictor

The input data of the model uses the characteristic data contained in the input indicators in Table 3, and the performance of 7 different classification methods is analyzed and compared to find the best prediction indicators for predicting course performance. This experiment selects 7 machine learning algorithms that are currently widely used in classification tasks in the learning and prediction field: SVC (R), SVC (L), Bayes, KNN (U), KNN (D), DT and Softmax. Select the commonly used indicators currently used to evaluate predictors, including accuracy (ACC), F1-score (F1), kappa (K) and computation time (Time).The three experimental groups adopt the method of five-fold cross-validation. Each data sets is divided into the same five groups, and four of them are used in a cycle as the training set, and the other is used as the test set for prediction. The experimental results use the mean of five-fold cross-validation, and the output value of the predictor is “qualified” or “unqualified”.

## 5. Experimental Discussion and Analysis

This section gives the results of the above experiments, including the accuracy (ACC), F1-score (F1), kappa (K) and computation time (Time) required for each experimental group under 7 machine learning methods. Through the analysis of these data mining effective behavior feature space, and verify the effectiveness of our feature fusion method.

### 5.1. Mining the Feature Space of Effective E-Learning Behavior

This paper designs an E-learning behavior feature set based on the EBC Model. In order to explore the impact of different behavior feature subsets on the prediction effect of learning performance, the performance of 7 different classification methods is used and compared. The performance of the feature subset under the seven algorithms is shown in Figure 5, Figure 6 and Figure 7. The experiment adopts the five-fold cross-validation method, and the values of accuracy, F1-score and kappa are the average values of the five-fold cross-validation. Figure 5, Figure 6 and Figure 7 show the maximum, upper quartile, mean (red in the figure), lower quartile, and minimum of the predicted effects of the seven algorithms.

In general, the three indicators all give similar experimental results. The accuracy of all feature subsets is in the range of 80.10–91.92%, F1-score is in the range of 0.8495–0.9445, and kappa is in the range of 0.4530–0.7960.In terms of the best performing feature subset, F10 obtained the best prediction effect on SVC (L), 91.92%, 0.9445, 0.7960, respectively, followed by F11 with 91.89%, 0.9443, 0.7949, respectively. In addition, F10, F11 performed best on averages under 7 algorithms, with 90.73%, 90.57% accuracy, and 0.9363, 0.9360 F1-score and 0.7655, 0.7569 kappa, respectively. Therefore, we choose F10 and F11 as the effective E-learning behavior feature space. Generally speaking, after feature selection, less important behaviors have been eliminated. F14 with all behavior characteristics should be the best predictor. However, the overall performance of F14 is not as good as F11. This may be due to social interaction behaviors. The forum discussion behavior included in CI has many distracting items, such as publishing forum information that is not related to learning, and deviating from the learning theme during discussion. In addition, questions in the forum do not require immediate answers, which may cause delays and affect learning. This kind of hard-to-control learning behavior may not be the best predictor. F11 includes basic interaction behavior, knowledge interaction behavior, and self-interaction behavior. F10 includes basic interaction behavior, knowledge interaction behavior and forum interaction behavior. In addition, F4 includes basic interaction behavior and knowledge interaction behavior. In online learning, basic interaction behaviors such as accessing platform pages and course pages, and knowledge interaction behaviors such as searching for resources and querying Wikipedia have a significant impact on E-learning performance. It can be seen that the human-computer interaction behavior in E-learning can better reflect the learning status of learners. Teachers should design questions suitable for different levels of learners to promote students’ active thinking and communication.

### 5.2. Validation of Feature Fusion

Based on the experimental results of Experiment 1, this paper fuses features of the effective E-learning behavior feature spaces F10 and F11 to improve the performance of the predictor. Three experimental groups are designed and 6 species (42 types) learning performance predictors are constructed based on 7 types of machine learning algorithms, and analyze whether the feature fusion strategy can improve the prediction effect according to the prediction effects of these predictors.In order to verify the stability of the algorithm, the three experimental groups all adopt the five-fold cross-validation method, and the values of accuracy, F1-score and kappa are all five. The average value of folded cross-validation, the results are shown in Figure 8, Figure 9 and Figure 10.

From Figure 8, Figure 9 and Figure 10 comparing the indicators of experimental groups 1, 2, and 3, it can be concluded that, in general, F10 and F11 have the best predictive effect in experimental group 3, which verifies the effectiveness of the feature fusion method.The feature subset F10 in group 3 shows the highest prediction performance, the accuracy, F1-score and kappa of SVC (R) are 98.44%, 0.9893 and 0.9600, respectively. The data of experimental group 1 without any processing obtained the worst experimental results. The average accuracy, F1-score, and kappa values of F10 and F11 under the 7 algorithms were 82.65%, 0.8809, 0.5624 and 81.95%, 0.8764, 0.5432, respectively. The accuracy of experimental group 2 was improved after feature selection, obtaining 90.73%, 0.9363, 0.7655 and 90.57%, 0.9360, 0.7569, respectively. The experimental group 3 uses the feature fusion strategy to further integrate the behaviors of the same category in the feature subset based on the idea of behavior classification, and the indicators under the 7 machine learning algorithms obtain the average accuracy, average F1-score, and average kappa values of F10 and F11 are 96.73%, 0.9775, 0.9177 and 96.89%, 0.9786, 0.9216, respectively. Compared with group 2, group 3 increased by 6.61%, 4.40%, 19.88% and 6.98%, 4.55%, 21.76%, respectively.

In order to comprehensively compare and analyze the prediction performance of the three groups of experiments, Table 5 gives the average ACC, F1-score, kappa and total computation time of the 15 feature subsets in group 1, group 2, and group 3 under 7 classification algorithms. The results in bold represent the best predicted performance for each row (that is, each algorithm) in the table. The results show that the prediction performance of group 2 after feature selection is significantly better than that of group 1, and group 3 further improves the prediction effect of group 2. Table 5 also presents the average performance of the 7 algorithms. Compared with group 1, the accuracy of group 2 is from 81.30% to 89.09%, the F1-score is from 0.872 to 0.925, and the kappa is from 0.523 to 0.724. Group 3 is further improved relative to Group 2, with an accuracy of 94.59%, F1-score of 0.963, and kappa of 0.862. In terms of total computation time, most methods obtained the shortest total computation time in group 3, which shows that the method proposed in this paper can effectively improve the computational speed of the model while optimizing the prediction effect.

It can be seen that the performance improvement strategy proposed in this paper has achieved excellent results in real scenarios. The use of feature fusion strategy to build a learning performance predictor can not only reduce the dimension of feature data, but also the prediction performance is significantly better than the traditional methods that only use data preprocessing or only use feature selection strategy on the basis of preprocessing to build a learning performance predictor.

### 5.3. Experimental Efficiency

In order to verify that the method proposed in this paper can effectively reduce the computational cost of the model, we recorded the average computation time of the three groups of prediction models for all input indicators, as shown in Figure 11. As can be seen from the figure, group 2 and group 3 are greatly improved compared to group 1. The average calculation time of F0–F14 of group 1 is in the interval of 0.0864 s to 0.1351 s, group 2 is in the interval of 0.0749 s to 0.0888 s, and group 3 is in the interval of 0.0657 s to 0.0813 s. The average of the total time required for the prediction of all feature subsets under each algorithm in the three groups of experiments is1.6450 s, 1.2194 s, 1.0862 s, respectively. The computational speed of group 2 is 25.87% higher than that of group 1, and the computational speed of group 3 is 10.92% higher than that of group 2. The results show that our work can optimize the model structure, reduce the complexity of the model, and increase the speed of computation.

## 6. Conclusions

Learning analysis aims to explore data-based learning diagnosis, prediction and intervention, which can help better understand and optimize the learning process and learning environment. In the context of solving the problem of predicting student test scores, this article attempts to find the most important behavior categories for learning performance prediction through different machine learning techniques from the perspective of behavior classification. This article proposes an E-learning behavior classification model (EBC Model) on the basis of summarizing existing research. The input data of the model considers the combination of behavior categories based on the EBC Model. In addition, different from traditional data processing methods, a new learning behavior feature fusion strategy is proposed, and its effectiveness is comprehensively analyzed and compared using machine learning technology. Through the data analysis of OULAD, this research digs out the effective E-learning behavior feature space as the basic interaction behavior and knowledge interaction behavior. Compared with interpersonal interaction (learners interact with peers, teachers, and themselves), human-computer interaction (learners interact with the system and learning content) has a greater impact on performance, which shows that online education does not have high requirements for students’ collaborative learning. The initiative of students to use the E-learning platform needs to be strengthened. Secondly, the experiment verified that the feature fusion strategy proposed in this paper reduces the dimension of feature data, can simplify the prediction model, extract more effective behavioral data for the model, and obtain better prediction results. This article made a new attempt based on the EBC Model, and the conclusions reached are slightly different from previous related studies. Most scholars believe that social interaction behaviors are significantly related to academic performance. The reason for the different conclusions may be that the density distribution of behavioral data in different data sets is different, and the measurement of student behavior is different, which affects the prediction effect of learning performance. At the same time, the research results show that the use of our data processing method can predict students’ course performance with high precision, and can provide a more reliable method for education practitioners to analyze the connection between educational phenomena and results.

Our future goal is to further optimize the EBC Model, consider multi-feature learner data, migrate the model to other E-learning platforms to build predictors, use the results of learning predictions to further build a learning diagnosis model, and carry out corresponding level interventions according to the level of prediction.

## Figures and Tables

**Figure 1 entropy-24-00722-f001:**
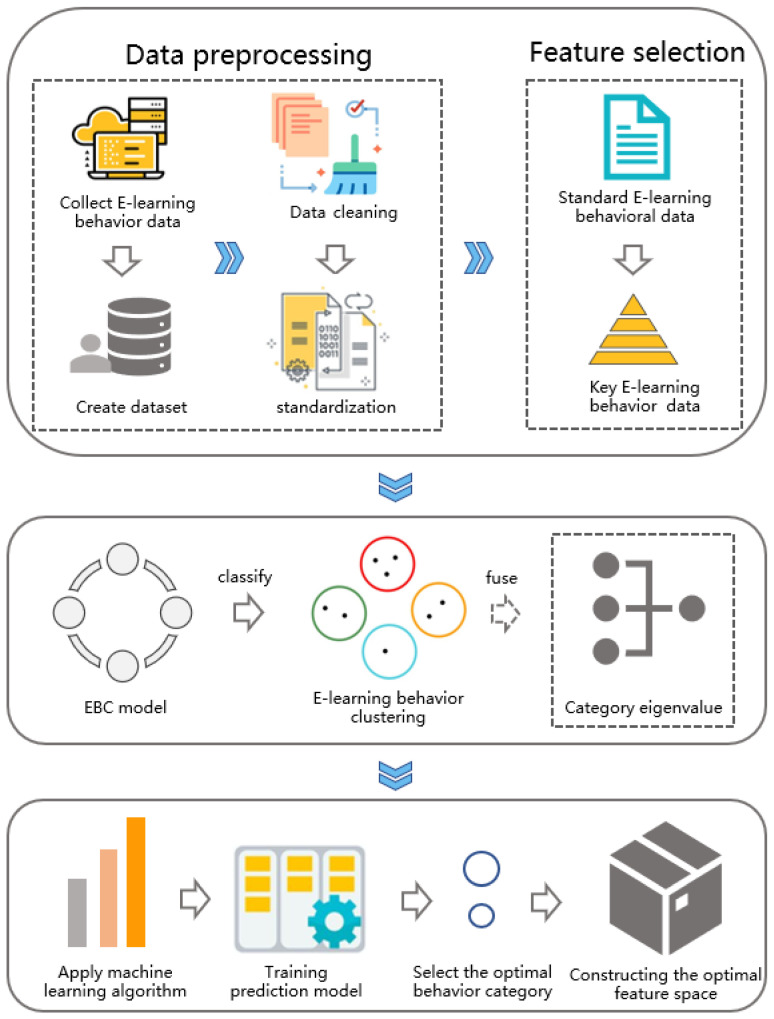
Framework of the proposed method.

**Figure 2 entropy-24-00722-f002:**
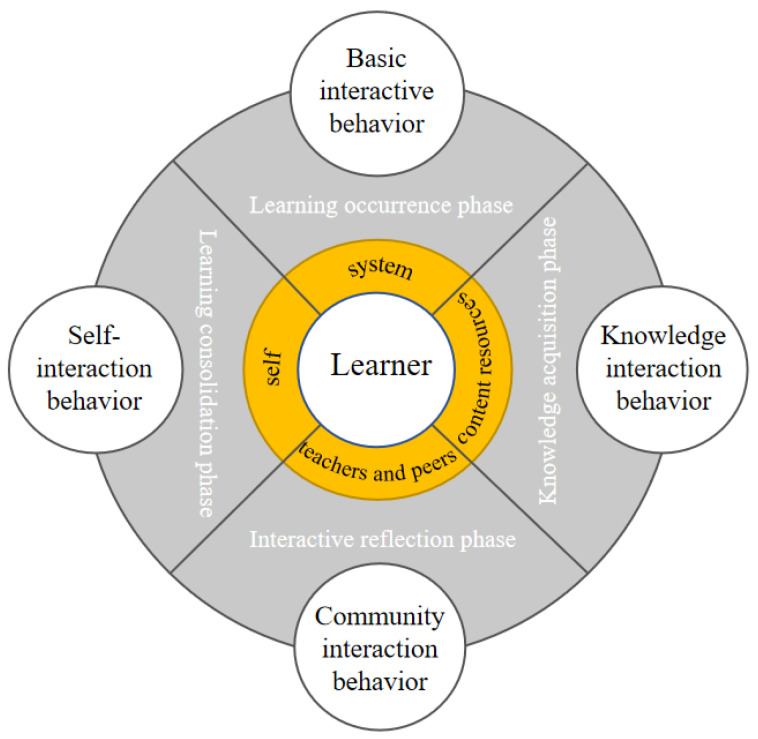
E-learning behavior classification model-EBC Model.

**Figure 3 entropy-24-00722-f003:**
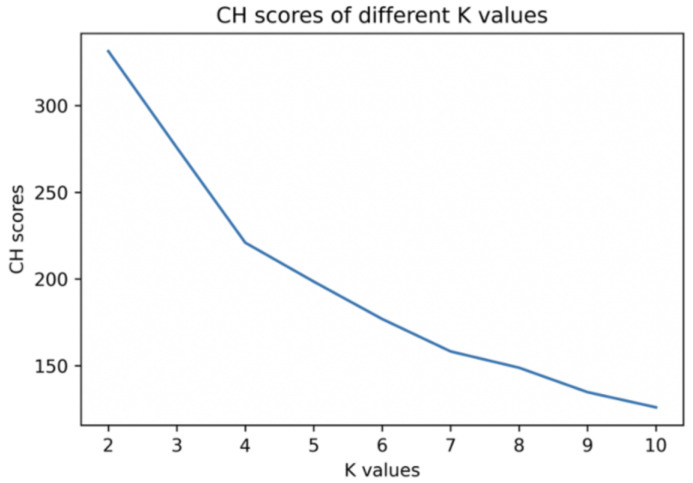
K kinds of clustering CH score chart.

**Figure 4 entropy-24-00722-f004:**
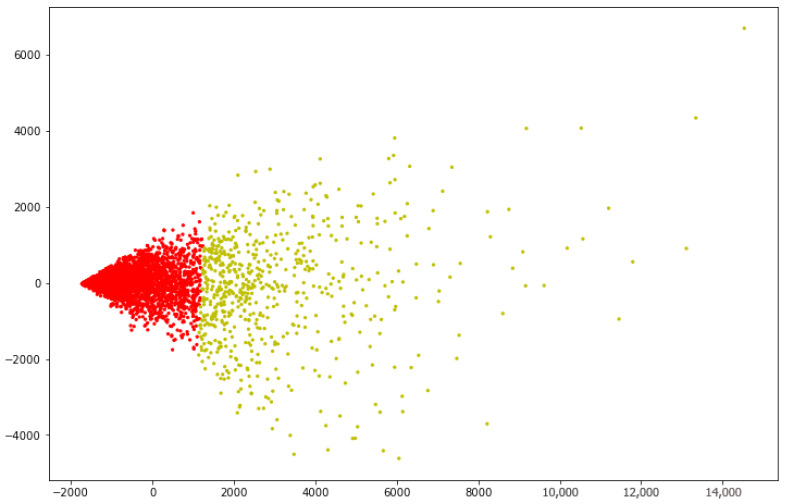
Visualizing clustering results.

**Figure 5 entropy-24-00722-f005:**
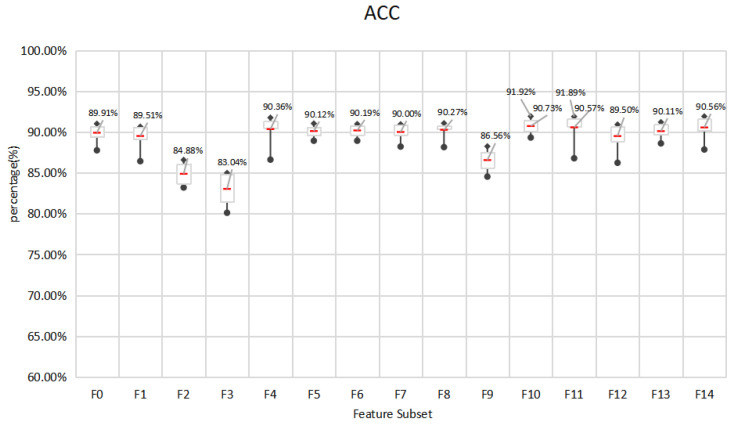
Accuracy of behavioral feature subsets under 7 algorithms.

**Figure 6 entropy-24-00722-f006:**
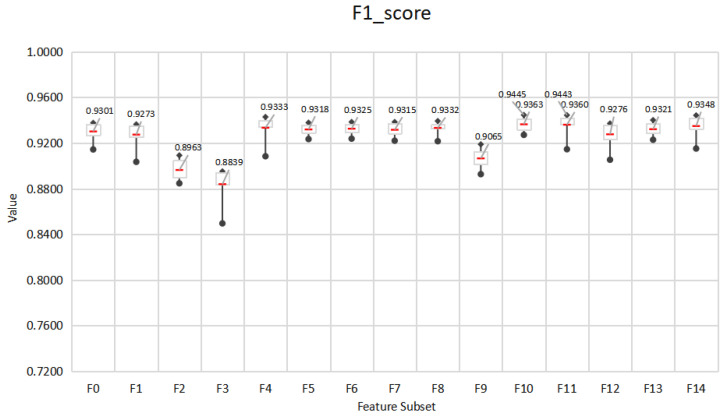
F1-score of behavioral feature subsets under 7 algorithms.

**Figure 7 entropy-24-00722-f007:**
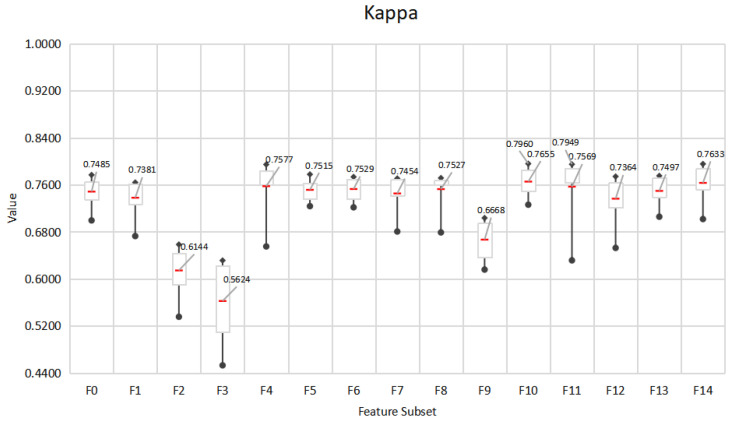
Kappa coefficients of behavioral feature subsets under 7 algorithms.

**Figure 8 entropy-24-00722-f008:**
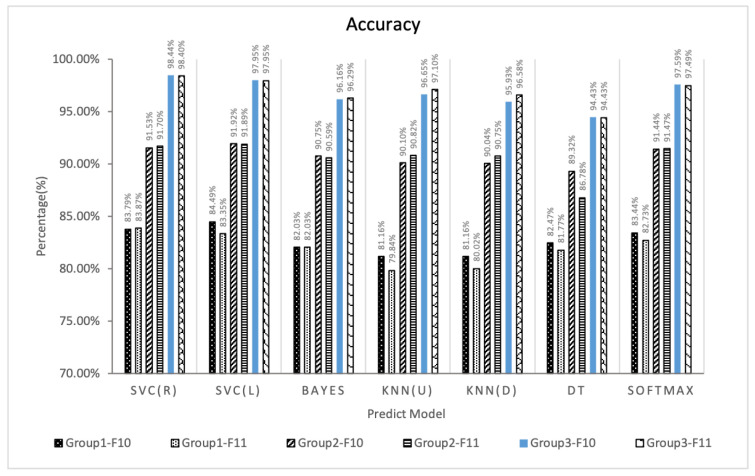
Accuracy of the three groups of prediction models.

**Figure 9 entropy-24-00722-f009:**
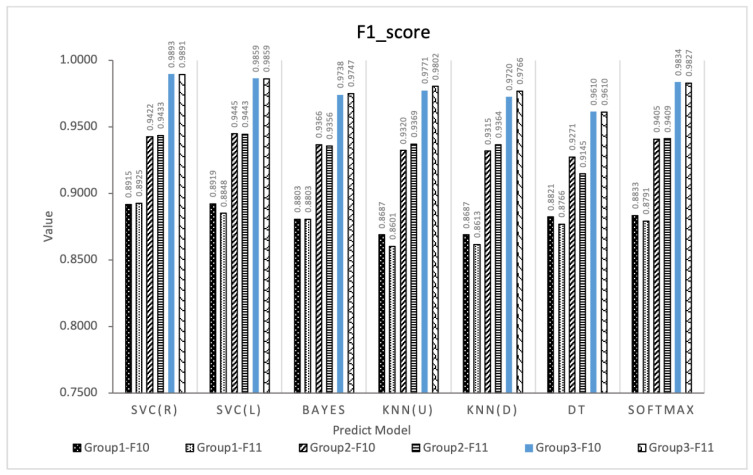
F1-score of the three groups of prediction models.

**Figure 10 entropy-24-00722-f010:**
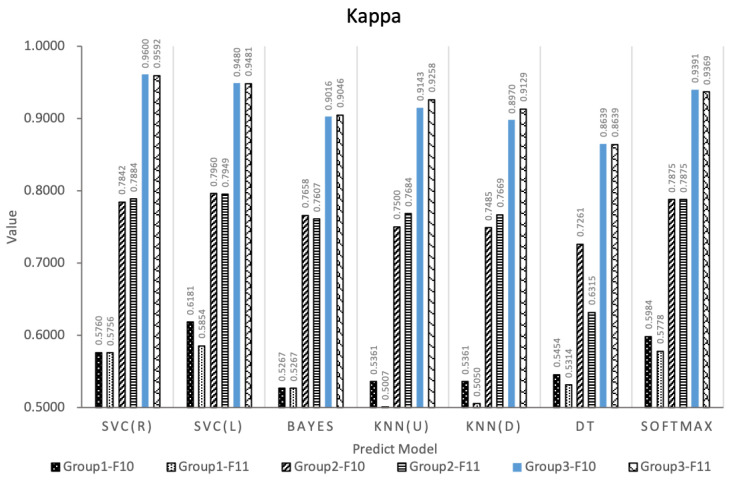
Kappa of the three groups of prediction models.

**Figure 11 entropy-24-00722-f011:**
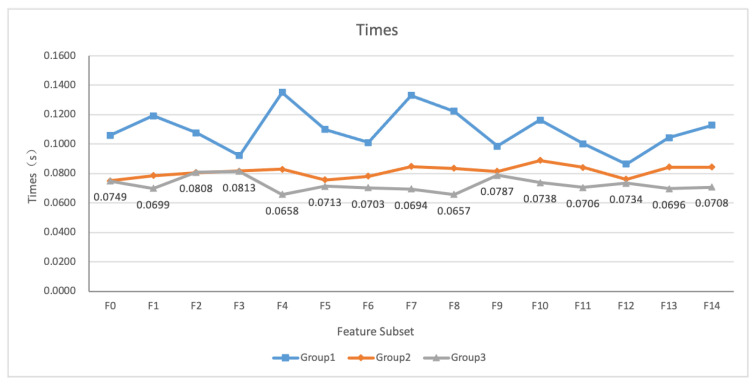
Computation time of the three groups of prediction models.

**Table 1 entropy-24-00722-t001:** Definitions of related symbols.

Symbols	Definition
$A\left\{a_{1},a_{2},\ldots ,a_{N}\right\}$	The original E-learning behavior sets *A*, $a_{n}$ is labels of E-learning behavior.
$p_{mn}$	Original E-learning behavior data, *n* represents the nth E-learning behavior, n=1,⋯,N, *m* represents the student user tag, m=1,⋯,M.
$p_{mn}^{\prime }$	Standardized E-learning behavior data.
$A^{\prime }\left\{a_{1}^{\prime },a_{2}^{\prime },\ldots \ldots ,a_{N}^{\prime }\right\}$	Standard E-learning behavior set $A^{\prime }$, $a_{n}^{\prime }$ is the label of standard E-learning behavior.
$S\left\{s_{1},s_{2},\ldots ,s_{N}\right\}$	Set of eigenvalues of learning behavior, $S_{n}$ represents the overall eigenvalue of the nth learning behavior.
$E_{n}\left\{a_{1}^{\prime },a_{3}^{\prime },\ldots ,a_{i}^{\prime }\right\}$	Learning behavior class $E_{n}$ is composed of $a_{1}^{\prime },a_{3}^{\prime },\ldots ,a_{i}^{\prime }$, is part of the standard E-learning behavior set $A^{\prime }$.
$S_{E}\left\{s_{E_{1}},s_{E_{2}},\ldots ,s_{E_{N}}\right\}$	E-learning behavior category feature value set, where $S_{E_{1}}$ represents the feature value after the feature fusion of behavior category $E_{1}$.

**Table 2 entropy-24-00722-t002:** E-learning behavior and coding of DDD courses.

Number	E-Learning Behavior	Behavior Interpretation	Behavior Coding	Behavior Category Coding
01	homepage	visit the homepage of the learning platform	H	BI
02	page	access the course page	P	BI
03	subpage	access the course subpage	S	BI
04	glossary	access glossary	G	KI
05	ouwiki	query with Wikipedia	W	KI
06	resource	search platform resources	R	KI
07	url	access course URL link	U	KI
08	oucontent	download platform resources	T	KI
09	forumng	participate in Forum discussion	F	CI
10	oucollaborate	participate in collaborative communication	C	CI
11	ouelluminate	participate in simulation seminars	E	CI
12	externalquiz	complete extracurricular quizzes	Q	SI

**Table 3 entropy-24-00722-t003:** E-learning behavior feature data after feature selection.

Method Feature & Feature Value	Entropy Feature	Feature Value	Variance Filtering Feature	Feature Value	Reserve
1	T (KI)	2.27 $\times 10^{-2}$	S (BI)	1.75 $\times 10^{6}$	✓
2	H (BI)	2.88 $\times 10^{-2}$	F (CI)	1.44 $\times 10^{6}$	✓
3	R (KI)	3.53 $\times 10^{-2}$	H (BI)	3.78 $\times 10^{5}$	✓
4	S (BI)	3.98 $\times 10^{-2}$	R (KI)	1.96 $\times 10^{5}$	✓
5	F (CI)	4.97 $\times 10^{-2}$	U (KI)	1.78 $\times 10^{5}$	✓
6	Q (SI)	5.40 $\times 10^{-2}$	T (KI)	5.93 $\times 10^{4}$	✓
7	U (KI)	6.52 $\times 10^{-2}$	W (KI)	2.43 $\times 10^{4}$	✓
8	W (KI)	6.70 $\times 10^{-2}$	Q (SI)	2.03 $\times 10^{4}$	✓
9	C (CI)	8.48 $\times 10^{-2}$	C (CI)	8.92 $\times 10^{3}$	✗
10	G (KI)	1.71 $\times 10^{-1}$	E (CI)	8.82 $\times 10^{3}$	✗
11	E (CI)	1.90 $\times 10^{-1}$	G (KI)	4.71 $\times 10^{3}$	✗
12	P (BI)	1.92 $\times 10^{-1}$	P (BI)	7.76 $\times 10^{1}$	✗

**Table 4 entropy-24-00722-t004:** E-learning behavior feature set.

Feature Subset	Behavior Category Coding	Behavior Coding
F0	BI	H, S
F1	KI	W, R, U, T
F2	CI	F
F3	SI	Q
F4	BI, KI	H, S, W, R, U, T
F5	BI, CI	H, S, F
F6	BI, SI	H, S, Q
F7	KI, CI	W, R, U, T, F
F8	KI, SI	W, R, U, T, Q
F9	CI, SI	F,Q
F10	BI, KI, CI	H, S, W, R, U, T, F
F11	BI, KI, SI	H, S, W, R, U, T, Q
F12	BI, CI, SI	H, S, F, Q
F13	KI, CI, SI	W, R, U, T, F, Q
F14	BI, KI, CI, SI	H, S, W, R, U, T, F, Q

**Table 5 entropy-24-00722-t005:** The average accuracy (ACC, %), F1-score (F1), kappa (K), and total computation time (T) of the three sets of prediction models.

Method	Group 1				Group 2				Group 3			
	ACC	F1	K	T	ACC	F1	K	T	ACC	F1	K	T
SVC (R)	83.17%	0.887	0.563	3.620	90.14%	0.932	0.751	4.497	**96.08%**	**0.973**	**0.899**	**3.982**
SVC (L)	81.65%	0.873	0.537	2.032	89.70%	0.927	0.749	1.852	**95.14%**	**0.966**	**0.882**	**1.573**
BAYES	81.67%	0.877	0.523	3.254	89.51%	0.928	0.735	**1.501**	**95.28%**	**0.968**	**0.877**	1.511
KNN (U)	80.20%	0.863	0.510	1.767	88.63%	0.923	0.706	0.449	**94.68%**	**0.964**	**0.863**	**0.390**
KNN (D)	79.92%	0.860	0.504	0.688	88.37%	0.921	0.701	0.121	**93.99%**	**0.959**	**0.846**	**0.066**
DT	80.90%	0.873	0.495	0.038	87.43%	0.914	0.677	0.018	**91.75%**	**0.944**	**0.786**	**0.016**
SOFTMAX	81.58%	0.873	0.532	0.116	89.82%	0.929	0.750	0.097	**95.19%**	**0.967**	**0.880**	**0.065**
AVE	81.30%	0.872	0.523	1.645	89.09%	0.925	0.724	1.219	**94.59%**	**0.963**	**0.862**	**1.086**

## Data Availability

The data is available in a publicly accessible database and can be found at https://analyse.kmi.open.ac.uk/open_dataset (accessed on 20 April 2022).

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
