# Peer review of "E-Learning Performance Prediction: Mining the Feature Space of Effective Learning Behavior"

_entropy, 2022, doi:10.3390/e24050722_

Round 1
Reviewer 1 Report
Contribution:
The main contribution of this paper is the new way of preprocessing MOOC's data into E-learning behavior feature space, application of a feature fusion algorithm, and building of classification models upon final feature representations.
The substantial advantage of the proposed method is the effective reduction of feature space and the absence of personal data usage, which could be unreachable in some setups. Also, a small ablation study conducted on feature usage could be considered as an advantage.
Problems:
1. Scalability
However, we question the scalability of this approach; there are no experiments conducted on datasets apart from the OULA dataset, which questions the performance of the proposed EBC model.
2. Feature Selection
The description of the feature selection approach according to their average standard deviations has left gaps in understanding of pathways that were used to obtain or select optimal thresholds for the value of Sn. Moreover, nothing was said even about the values of thresholds.
3. Validation
The presented results on the OULA dataset itself seem vague due to the absence of mention of any cross-validation techniques (line 523). It is worth noting that the OULA dataset contains time-series data for several iterations of one course, which means that per-course-iteration validation is required in order not to exploit test data with already known future train data observations.
4. Novelty
Also, this paper does not mention any other authors' efforts on the OULA dataset. However, there is plenty of work solving the “at-risk students identification” problem on this dataset. Hosta et al. [1] are considering a more complicated setup without exploiting whole-course activity knowledge; authors of [2, 3] are considering exactly the same setup (aggregating the whole course info for one student) and are demonstrating comparable or even better results with direct usage of VLE data, which questions novelty and usefulness of the proposed method.
5. Ambiguous Statements
This paper is full of vague statements: “...use of multi-dimensional data may affect the generalization ability…” (line 161) - in [2, 3] multi-dimensional data usage does not affect performance; “...it is necessary to use feature extraction methods…” (line 259) - at the same time, Haiyang et al. [4] successfully used a NN-based approach to predict performance from time-series and achieved accuracy over 94%; etc.
6. Writing Mistakes
There are a lot of misprints and writing mistakes in this article. Here we present some of them: equation (6) - m letter in the denominator in the right part of the equation probably should be uppercase because according to the information below “m = 1,···,M” (line 421) where M is the number of students; “and self-interaction behaviors is good” (line 449) here should be used “are” instead of “is” because of narration rules; “Predictor.” (line 482) - sentence which contains only one word; and so on.
7. Self-plagiarism
The very same research was already published in Nature Scientific Reports:
https://www.nature.com/articles/s41598-021-03867-8
Despite it not exactly the same text, we have no doubts that these two texts describe the very same research.
Summary:
There is no proper overview of related work. The credibility of the proposed method results is unclear due to the absence of cross-validation. The scientific novelty is not clear as there are many similar papers with similar results and another paper of the same authors already published.
References:
- Hlosta, M., Zdrahal, Z., & Zendulka, J. (2017, March). Ouroboros: early identification of at-risk students without models based on legacy data. In Proceedings of the seventh international learning analytics & knowledge conference (pp. 6-15).
- Hussain, M., Zhu, W., Zhang, W., & Abidi, S. M. R. (2018). Student engagement predictions in an e-learning system and their impact on student course assessment scores. Computational intelligence and neuroscience, 2018.
- Adnan, M., Habib, A., Ashraf, J., Mussadiq, S., Raza, A. A., Abid, M., ... & Khan, S. U. (2021). Predicting at-risk students at different percentages of course length for early intervention using machine learning models. IEEE Access, 9, 7519-7539.
- Haiyang, L., Wang, Z., Benachour, P., & Tubman, P. (2018, July). A time series classification method for behaviour-based dropout prediction. In 2018 IEEE 18th international conference on advanced learning technologies (ICALT) (pp. 191-195). IEEE.
Author Response
Q1: Scalability
However, we question the scalability of this approach; there are no experiments conducted on datasets apart from the OULA dataset, which questions the performance of the proposed EBC model.
Your comments are very important, and the research goal of this paper is mainly to prove the validity of the EBC model and to further mine the valid data subsets distinguished by the EBC model.
- First, perform an ablation experiment on the effective behavioral feature subset (group2). Finally, it is concluded that the feature subset F10 has the highest accuracy of 91.65% under 7 algorithms. F11 has the strongest stability among the 7 algorithms, with the highest average accuracy of 91.06%. For the data, please refer to Figure 4-6 of the manuscript. This work has mined effective feature subsets.
- Secondly, in terms of the validity of the EBC model (group 3), the ACC, F1 and Kappa average indicators of the seven machine learning algorithms predicted by the EBC model have been further improved. Respectively, the ACC average increased from 89.13% to 92.74%, the F1 average increased from 92.39% to 94.91%, and the Kappa average increased from 73.34% to 82.26%. Please refer to Table 5 of the manuscript for the data. This work verifies that the proposed EBC model can effectively improve the prediction accuracy.
- Finally, the effective feature subsets F10 and F11 are used to predict by feature fusion (group3). The KNN algorithm based on the data subset F10 has the best performance, the highest value of ACC is 96.10%, the highest value of F1 is 97.28%, and the highest value of Kappa is 90.43%. Please see Figures 7-9 of the manuscript for the data. This work proves that our proposed method can further improve the prediction effect.
The reviewer's comments are very meaningful, and we will further work on the scalability of our method in the future to validate it on more datasets.
Q2: Feature Selection
The description of the feature selection approach according to their average standard deviations has left gaps in understanding of pathways that were used to obtain or select optimal thresholds for the value of Sn. Moreover, nothing was said even about the values of thresholds.
Thank you for your valuable comments. Regarding the optimal threshold of Sn value, it is not clearly described. Now, the description has been added to the line 482-488 in the text and marked in red. We carried out double analysis by entropy weight method and variance filtering method, and the results were basically the same. The principles for selecting features are: 1. In order to ensure the diversity of features, at least one feature is reserved under each behavior class; 2. Feature compression is performed on each class as much as possible. According to the above principles, we finally select the top 8 features for the results of the entropy weight method and the variance filtering value in the table below. For better description, table 3 in the manuscript is revised as shown in the table below.
Q3: Validation
The presented results on the OULA dataset itself seem vague due to the absence of mention of any cross-validation techniques (line 523). It is worth noting that the OULA dataset contains time-series data for several iterations of one course, which means that per-course-iteration validation is required in order not to exploit test data with already known future train data observations.
The question you mentioned about time series data is very meaningful. The main research goal of this paper is to mine the effective feature space. In order to ensure the comprehensiveness of the feature space of the data, the experimental design of this paper does not consider the characteristics of time series, but mines the full time series data after the course. Because this paper aims to mine the feature space, we hope to illustrate the effectiveness of different feature spaces by the difference of the prediction results. The cross-validation experimental technique can better solve the problem of data randomness, but it will not affect the experimental results of this paper too much. Because the revision time is tight, we have started the experiment of 5-fold cross-validation. If necessary, we can submit it in the next revision.
Q4: Novelty
Also, this paper does not mention any other authors' efforts on the OULA dataset. However, there is plenty of work solving the “at-risk students identification” problem on this dataset. Hosta et al. [1] are considering a more complicated setup without exploiting whole-course activity knowledge; authors of [2, 3] are considering exactly the same setup (aggregating the whole course info for one student) and are demonstrating comparable or even better results with direct usage of VLE data, which questions novelty and usefulness of the proposed method.
Thank you for your very meaningful suggestion, Hosta et al. [1] aims to solve the problem of data imbalance, his method can be very good for predicting students at risk. The purpose of this paper is to mine the effective feature space. Mining the effective feature space for behavior data and obtaining the feature space with dimensionality reduction has certain reference significance for related research work in this field.
Regarding the experimental results of Hussain et al. [2], the highest accuracy rate is 94.7%, while the experimental results studied in this paper are: the average accuracy rate is above 92%, and the highest accuracy rate of KNN based on the F10 data subset is 96.10%, see Figure 7 of the manuscript for data.
The experimental results of Adnan et al. [3] have the highest prediction rate of 92%, and the experimental results of this study are: the average accuracy rate is above 92%, and the highest accuracy rate of KNN based on the F10 data subset is 96.10%, see Figure 7 of the manuscript for data.
The original Table 5 only gives the average accuracy of each feature subset. For a clearer presentation, we have added the average of 7 algorithm metrics to Table 5, which has been revised in the latest submitted manuscript. The purpose of this paper is to mine the effective feature space. The average and highest accuracy data are not the only criteria for judging the innovativeness of our work. The main purpose is to judge the validity of the data space we have excavated. I believe our work still has certain reference significance.
Q5: Ambiguous Statements
This paper is full of vague statements: “...use of multi-dimensional data may affect the generalization ability…” (line 161) - in [2, 3] multi-dimensional data usage does not affect performance; “...it is necessary to use feature extraction methods…” (line 259) - at the same time, Haiyang et al. [4] successfully used a NN-based approach to predict performance from time-series and achieved accuracy over 94%; etc.
Thank you for your important suggestion. Regarding line161, we believe that the use of multi-dimensional data may affect the generalization ability of the prediction model, which is also the purpose of this article, mining and reducing the dimensionality of the data space without affecting the performance of the algorithm.
Regarding line259, Hussain et al [2] and Adnan et al [3] mentioned that multi-dimensional data will not affect performance, but our experiments have proved that data of reasonable dimensions is conducive to improving the performance of the algorithm. Please refer to Table 5 for the results.
Regarding the experimental results of Liu et al., the highest prediction rate is 94%; the average accuracy rate is above 92%, and the highest accuracy rate of KNN based on the F10 data subset is 96.10%. Please refer to Figure 7 of the manuscript for the data.
The original Table 5 only gives the average accuracy of each feature subset. For a clearer presentation, we have added the average of 7 algorithm metrics to Table 5, which has been revised in the latest submitted manuscript. The purpose of this paper is to mine the effective feature space. The average and highest accuracy data are not the only criteria for judging the innovativeness of our work. The main purpose is to judge the validity of the data space we have excavated. I believe our work still has certain reference significance.
Q6: Writing Mistakes
There are a lot of misprints and writing mistakes in this article. Here we present some of them: equation (6) - m letter in the denominator in the right part of the equation probably should be uppercase because according to the information below “m = 1,···,M” (line 421) where M is the number of students; “and self-interaction behaviors is good” (line 449) here should be used “are” instead of “is” because of narration rules; “Predictor.” (line 482) - sentence which contains only one word; and so on.
Thank you for your valuable suggestion. There is indeed a writing mistake in formula (6), which has now been revised in the manuscript as follows: (screenshot + M changed to m, the m below is no problem).
Line421 is ambiguous because of a writing mistake in formula 6. The formula has been revised and the description is correct.
Writing errors in Line449 and Line482 have been corrected in the manuscript.
We have carefully scrutinized the manuscript, and made corresponding revisions including some typos, grammatical errors and long sentences, etc.
Q7: Self-plagiarism
The very same research was already published in Nature Scientific Reports:
https://www.nature.com/articles/s41598-021-03867-8
Despite it not exactly the same text, we have no doubts that these two texts describe the very same research.
We understand the reviewer's concerns, and for that article published in Scientific Reports, our focus is to propose a learning performance prediction framework based on online behavior classification and verify its effectiveness. Our submission to your journal is a continuation of the Scientific Reports article. First, we classify and combine behavioral features, build a feature space, mine more effective behavioral features, and further improve the performance of the predictor. For your concerns, we detected similarity between two articles on the Copyleaks platform. Copyleaks is one of the best plagiarism software detection tools on the market, Copyleaks can compare text online and perform extensive searches on the Internet and various databases to find similar content, showing only relevant results. The repetition rate is 4.5%, as follows:
We sincerely hope that this revised manuscript can resolve all your comments and suggestions. We thank the reviewers for their enthusiastic work and hope that our revision would be approved. Again, thank you very much for your comments and suggestions.
Yours sincerely,

Reviewer 2 Report
The article is devoted to an important study of students' behavior in distance learning and invesgation a knowledge prediction. The authors identify different features and using different known methods to investigate the effectiveness of these features. Based on these studies, a conclusion is made about predicting the effectiveness of E-learning.
Remarks:
- The authors do not provide evidence that their chosen features fully cover all behavioral characteristics. Therefore, in the title of the article the word "optimal" behavior in teaching is used only in relation to the introduced features of the authors, and not to the problem as a whole, so I would recommend replacing the word "optimal" in the title with "effective" or " expedient" as more correct.
- In distance learning, an important point is the objectivity of the knowledge test data, in particular, whether it is really the student's own answers or someone who helped him from outsiders, which was not controlled. Such parameters were not studied in the work.
- The "computational time" indicator is not important for this problem. As shown in this article, "the full prediction for each algorithm is 3.5892 s, 2.7294 s and 2.5472 s, respectively." This time is determined by the computing resources used and is not an objective indicator. Moreover, this is not a real-time problem.
- The results of E-learning for specialized learning this is when students are united by the study of specialized subjects (only natural sciences or only humanities) have not been studied. Then other results are possible, different from those given in this article.
- In distance learning, the important point is the objectivity of the knowledge test data, in particular whether it is really the student's own answers or someone who helped him from outside, which is not controlled.
- Not all abbreviations are decrypted, for example 247 lines. All abbreviations should be explained throughout the text.
- The formula (line 327) is not clarified relative to the index of small n. (line 329) relative to b usually such an index does not denote integers. Similarly, the letter x is usually used to define unknown quantities and not integers (lines 332, 333). In the designations E1 S it small or big letter? Similar in line 335. Where correct - in this row or in Table 1?
You need to correct the symbols throughout the text of the article. - In formula (1) and below (lines 402,403) different symbols and the same variables - straight and inclined?
You need to correct the formulas throughout the text of the article.
Author Response
Response for Reviewer 2:
Q1: The authors do not provide evidence that their chosen features fully cover all behavioral characteristics. Therefore, in the title of the article, the word "optimal" behavior in teaching is used only in relation to the introduced features of the authors, and not to the problem as a whole, so I would recommend replacing the word "optimal" in the title with "effective" or " expedient" as more correct.
Thank you very much for your suggestion, our behavioral features are selected from the DDD subjects of STEM courses in the OULA dataset, so not all behavioral features are covered, thank you for your constructive comments, we have changed "optimal" in the full text to "effective".
Q2: In distance learning, an important point is the objectivity of the knowledge test data, in particular, whether it is really the student's own answers or someone who helped him from outsiders, which was not controlled. Such parameters were not studied in the work.
Thanks to the reviewer for your profound suggestion. The issues you consider are also current research hotspots. When students study and take tests, it is very important to monitor students' violations. However, such an analysis is beyond the scope of our paper, which seeks to find more effective learned behavioral features in specific datasets and to improve prediction performance.
Considering the costs involved, and also not significantly supporting the thesis of this paper, this suggestion you make will be actively considered in our future work.
Q3: The "computational time" indicator is not important for this problem. As shown in this article, "the full prediction for each algorithm is 3.5892 s, 2.7294 s and 2.5472 s, respectively." This time is determined by the computing resources used and is not an objective indicator. Moreover, this is not a real-time problem.
We agree with your point of view that the problem discussed in this paper is not a real-time problem and the operation time is not particularly important, but we just want to make a record description of the running time of the overall algorithm experiment. It may be shown that our method improves some computational efficiency, and we hope that future work can be more efficient on this basis. For this reason, we have decided not to make this cut for now.
Q4: The results of E-learning for specialized learning this is when students are united by the study of specialized subjects (only natural sciences or only humanities) have not been studied. Then other results are possible, different from those given in this article.
Thanks to the reviewers for their constructive comments, the research in this paper is mainly based on the OULA data set, which is mainly based on natural sciences. The results of this research have laid a good foundation for the study of learning analysis. Nonetheless, this dataset is limited by the lack of a good separation of the natural and humanities disciplines. Since the results of this study have provided good and important reference results and foundations, future research can continue to explore various disciplines based on the results of this research. In future work, we will focus on distinguishing outcome studies from the natural and humanities.
Q5: In distance learning, the important point is the objectivity of the knowledge test data, in particular whether it is really the student's own answers or someone who helped him from outside, which is not controlled.
We thank the reviewers for their profound suggestions, such as the reply to question 2. The issues you consider are also current research hotspots. When students study and take tests, it is very critical to monitor students' violations. However, such an analysis is beyond the scope of our paper, which seeks to find more effective learned behavioral features in specific datasets and to improve prediction performance. Considering the costs involved, extending our dataset is neither feasible nor significantly supports our thesis, and this suggestion you make will be actively considered in our future work.
Q6: Not all abbreviations are decrypted, for example 247 lines. All abbreviations should be explained throughout the text.
Thanks for your constructive suggestion, which is highly appreciated. We have made modifications to the manuscript and marked in red.
Q7: The formula (line 327) is not clarified relative to the index of small n. (line 329) relative to b usually such an index does not denote integers. Similarly, the letter x is usually used to define unknown quantities and not integers (lines 332, 333). In the designations E1 S it small or big letter? Similar in line 335. Where correct - in this row or in Table 1?
You need to correct the symbols throughout the text of the article.
line 327, we have changed S to s and marked in red in the manuscript.
line 329, we have changed b to N and marked in red in the manuscript.
line332, 333, we have changed x to i and marked in red in the manuscript.
In addition to this, we have re-checked the entire manuscript, where editorial errors have been revised and marked in red in the manuscript.
Q8: In formula (1) and below (lines 402,403) different symbols and the same variables - straight and inclined?
line 402, 403, we rechecked and have changed variables and formulas to italics.
You need to correct the formulas throughout the text of the article.
The comments of the reviewers are greatly appreciated, we have re-checked the entire manuscript, and where there are editorial errors have been revised and marked in red in the manuscript.
We sincerely hope that this revised manuscript can resolve all your comments and suggestions. We thank the reviewers for their enthusiastic work and hope that our revision would be approved. Again, thank you very much for your comments and suggestions.
Yours sincerely,

Reviewer 3 Report
It is a well-building paper, without flaws.
This article about Learning Analysis provides us a very interesting model for predicting academic performance in e-learning.
This new learning behaviour model is support for a deep statistical analysis and show some practical suggestions for online learners and managers.
Congratulations.
Author Response
Response for Reviewer 3:
It is a well-building paper, without flaws.
This article about Learning Analysis provides us a very interesting model for predicting academic performance in e-learning.
This new learning behaviour model is support for a deep statistical analysis and show some practical suggestions for online learners and managers.
Congratulations.
Thank you very much for the encouraging recommendation.
We are happy to submit the revised manuscript to you. We thank the reviewers for their careful review of our manuscript and for their very helpful comments and suggestions to guide our revisions, and we are pleased that the editors and reviewers found our work to be meaningful.
Yours sincerely

Round 2
Reviewer 2 Report
Thanks to the authors for taking into account all my comments.
There are technical errors that need to be removed in the final version of the article:
1. lines 483, 488 insteed table need Table
2. line 488 insteed e-learning need E-learning
3. line 565 word Snown need delete
4. May be the technical remarcs else present. I recommende yet time more detail to check your article
I recommend to accept paper after minor revision - text editing
Author Response
Dear Reviewer,
Thank you very much for taking the time to review this manuscript. Thank you for your recognition of our work and your valuable suggestions, we have made corresponding changes and blue flags to your questions. And we reviewed the full text again to avoid writing errors. Please find our item-by-item responses below and find my corrections in the resubmitted file. Thank you again for carefully reviewing our manuscript and providing very useful comments and suggestions to guide our revisions.
There are technical errors that need to be removed in the final version of the article:
Point1. lines 483, 488 insteed table need Table.
Response 1: In lines 483, 488 we have modified table to Table.
Point2. line 488 insteed e-learning need E-learning.
Response 2: In line 488, we have modified e-learning to E-learning.
Point3. line 565 word Snown need delete.
Response 3: In line 565, we have deleted word Snown.
Point4. May be the technical remarcs else present. I recommende yet time more detail to check your article.
Response 4: Thanks for your suggestion, we have carefully examined the manuscript and corrected some errors.
Best regard,
Guodao Zhang
